# A Cell Double-Barcoding System for Quantitative Evaluation of Primary Tumors and Metastasis in Animals That Uncovers Clonal-Specific Anti-Cancer Drug Effects

**DOI:** 10.3390/cancers14061381

**Published:** 2022-03-08

**Authors:** Arkadi Hesin, Santosh Kumar, Valid Gahramanov, Maria Becker, Maria Vilenchik, Ilya Alexandrov, Julia Yaglom, Michael Y. Sherman

**Affiliations:** 1Department of Molecular Biology, Ariel University, Ariel 40700, Israel; arkadihe@ariel.ac.il (A.H.); kumars@ariel.ac.il (S.K.); validg@ariel.ac.il (V.G.); juliaya@ariel.ac.il (J.Y.); 2Adelson School of Medicine, Ariel University, Ariel 40700, Israel; mariabe@ariel.ac.il; 3Felicitex Therapeutics, Inc., Natick, MA 01760, USA; mvilenchik@felicitex.com; 4ActivSignal, Inc., Natick, MA 01760, USA; ialexandrov@activsignal.com

**Keywords:** preclinical studies, cell barcoding, metastasis, drug effects, xenograft

## Abstract

**Simple Summary:**

The main problem in treating advanced cancers is a metastatic spread when individual cancer cells leave the primary tumor and colonize to distant organs. In drug development, it is important to quantitatively assess effects of novel drug candidates on both primary tumors and metastasis. Unfortunately, current methods of monitoring metastasis in mouse models have low sensitivity and are not quantitative. Here, we developed a methodology to monitor drug effects on metastasis that is quantitative and has a very high sensitivity and resolution. In fact, it allows monitoring effects of drugs on individual cancer cells in animals.

**Abstract:**

Imaging in monitoring metastasis in mouse models has low sensitivity and is not quantitative. Cell DNA barcoding, demonstrating high sensitivity and resolution, allows monitoring effects of drugs on the number of tumor and metastatic clones. However, this technology is not suitable for comparison of sizes of metastatic clones in different animals, for example, drug treated and untreated, due to high biological and technical variability upon tumor and metastatic growth and isolation of barcodes from tissue DNA. However, both numbers of clones and their sizes are critical parameters for analysis of drug effects. Here we developed a modification of the barcoding approach for monitoring drug effects on tumors and metastasis that is quantitative, highly sensitive and highly reproducible. This novel cell double-barcoding system allows simultaneously following the fate of two or more cell variants or cell lines in xenograft models in vivo, and also following the fates of individual clones within each of these populations. This system allows comparing effects of drugs on different cell populations and thus normalizing drug effects by drug-resistant lines, which corrects for both biological and technical variabilities and significantly increases the reproducibility of results. Using this barcoding system, we uncovered that effects of a novel DYRK1B kinase inhibitor FX9847 on primary tumors and metastasis is clone-dependent, while a distinct drug osimertinib demonstrated clone-independent effects on cancer cell populations. Overall, a cell double-barcoding approach can significantly enrich our understanding of drug effects in basic research and preclinical studies.

## 1. Introduction

Quantitative evaluation of drug effects on tumor growth and metastasis in xenograft animal models is critical for basic understanding of tumor dynamics and for anti-cancer drug development. Currently used methods that are mainly based on either imaging or direct measurements of tumor and metastasis sizes have critical drawbacks: (a) they are semi-quantitative, thus requiring relatively large animal group sizes to achieve statistically significant data; (b) sensitivity of these methods is insufficient for detecting small metastasis. Indeed, in many cases primary xenograft tumors formed by human cancer cells injected subcutaneously in nude mice do not generate detectable metastasis. To obtain such metastasis, injection of cells into orthotopic organs or the tail vein is done. These approaches, however, are elaborate and CROs rarely use such methods in the preclinical studies, especially since the U.S. Food and Drug Administration (FDA) accepts preclinical data using subcutaneous xenografts (see, for example, [1,2,3]). When metastasis develops in animals, detection by imaging reflects a combination of the number of metastases and their individual sizes without distinguishing these parameters, which is very important.

Cell DNA barcoding allows dissecting many aspects of tumor dynamics, metastasis and drug effects [4,5,6,7,8,9,10,11,12,13,14]. Systems for cell barcoding include direct introduction of barcodes via lentiviral infection, CRISPR-based barcoding, or a combination of barcoding with fluorescent markers [9,10,11,14,15,16,17]. A very important advantage of these systems is high sensitivity and the ability to simultaneously monitor the fate of thousands and even millions of cells [5,6,18]. High sensitivity allows fine dissection of the metastatic process in xenograft models, cell lines and PDX models [5,8,11,17,19,20].

Despite these advantages, the barcoding approach has not been effectively used for quantitative evaluation of drugs’ effects. Though there were reports in which drug effects were studied using cell barcoding, most of these studies were done in vitro [5,6,10,21,22], where isolation of DNA sufficiently pure for the barcode amplification is a lesser problem. In several cases, when barcoding was used to assess drug effects in vivo, effects on the number of clones that formed primary tumors and metastasis were reported [8,20,23]. However, comparing sizes of individual clones that formed tumors and metastasis in drug-treated and untreated animals is problematic for regular cell barcoding. This drawback is because PCR-based isolation of barcodes shows high variability due to the differences in tissue disruption and purification of DNA, thus precluding effective comparison between animals of the number of reads belonging to each barcode (reflecting size of each clone). Importantly, assessment of the number of clones without assessing clone sizes is insufficient for evaluation of drug effects (see an example with osimertinib below). This problem was encountered, for example, in a published study of the effects of cisplatin in a PDX model measured by standard barcoding [20]. Cisplatin had a minor impact on clonal diversity while strongly reducing the tumor size, indicating that standard barcoding was unable to demonstrate the drug effect.

To circumvent these drawbacks, we developed a double-barcoding system that significantly improves measurements of primary tumors and metastasis. Our system enables quantitative measurements due to internal normalization of drug effects in individual animals that corrects for the variability in barcode isolation and significantly increases the reproducibility of results. Reduced variability in barcode isolation permits reducing number of animals per group and allows multiplexing to monitor cancer parameters in several cell lines simultaneously. Using this system, we uncovered that some drugs suppress growth of the entire cancer cell population, while others kill a subset of clones while not affecting other clonal subpopulations.

## 2. Materials and Methods

### 2.1. Cell Culture and Reagents

All cell lines were obtained from the ATCC (Manassas, VA, USA). The 4T1 (RRID:CVCL_0125) and H1975 (RRID:CVCL_1511) cell lines were cultured in RPMI-1640 medium supplemented with 2 mM L-Glutamine, 1% penicillin/streptomycin (pen/strep) and 10% *v*/*v* Fetal Bovine Serum (FBS); A549 (RRID:CVCL_0023) cells were cultured in Ham’s F12 Nutrient Medium supplemented with 2 mM L-Glutamine, 1% pen/strep and 10% FBS; MDA-MB-231 (RRID:CVCL_0062) and HEK293T (RRID:CVCL_0063) cell lines were cultured in DMEM, High Glucose (4500 mg/L D-glucose) supplemented with 2 mM L-Glutamine, 1% pen/strep and 10% FBS and HCT 116 cells were cultured in McCoy’s 5A Medium supplemented with 2 mM L-Glutamine, 1% pen/strep and 10% FBS. All the cell culture media and supplements including FBS were purchased from Biological Industries (Kibbutz Beit-Haemek, Kibbutz, Israel).

Cells were routinely cultured in 75 cm^2^ tissue culture flask and kept in a humidified atmosphere with 5% CO_2_ at 37 °C. All cell lines were tested for mycoplasma contamination every 3 months using commercial PCR-based kit (Biological Industries).

FX9847 was provided by Felicitex Therapeutics, Inc. (Natick, MA, USA); Osimertinib (Cat#O-7200) was purchased from LC Laboratories (Woburn, MA, USA); Irinotecan (Cat# I1406), 5-Azacytidine (Cat#A2385) were purchased from Sigma-Aldrich (St. Louis, MO, USA).

### 2.2. Single Cell Cloning

Colorectal cancer cell line HCT116 was used as model system; single cell cloning was performed by using limiting serial dilution and cloning disks (Merck, Darmstadt, Germany, Cat#Z374431).

### 2.3. Colony Growth Assay

H1975 and A549 cells were plated at 6-well plates at confluency of 50% and a day later treated with elevated concentrations of osimertinib or with vehicle control. The treatments were 2.5, 5, 7.5, 10, 12.5, 15, 20 μM osimertinib for 7 days, after which medium was removed and cells were rinsed with PBS. Fixation and staining of clones were done with a mixture of 0.5% crystal violet in 50/50 methanol/water for 30 min. Dishes were rinsed with water and left for drying at room temperature. Afterwards, the stain was solubilized in 0.2% Triton X-100 and the O.D.at 590 nm was determined [24].

### 2.4. Construction of the Double Barcode Library

A total of 24 barcoding libraries were constructed. Each library was obtained by cloning an oligonucleotide containing 8nt barcode, unique for each of the 24 libraries, followed by a 20 nt sequence common for all libraries, and followed by a 14 nt random sequence. On both sides the sequences of the oligoes were flanked by restriction sites. They were cloned into the lentiviral vector, and bacterial cells were transformed by electroporation to achieve high rate of transformation, in order to obtain highly representative libraries. We estimate that libraries contained 40,000–200,000 clones, and therefore similar numbers of diverse random sequences. Accordingly, they allow following the fates of about this number of individual clones.

In cell culture experiments, which did not require multiplexing, we barcoded cells with 50 M library from Cellecta Inc. (Mountain View, CA, USA), according to the manufacturer’s protocols (Cat#BC13X13-P). These barcodes contain 18-nucleotide sequence, which enables tagging individual cells with unique barcodes.

### 2.5. Virus Preparation for the Barcoding Library

Lentivirus barcoding library was prepared using the corresponding plasmids along with virus coat packaging plasmids using Lipofectamine 3000 infection reagent (Cat# L3000015, Thermo Scientific, Waltham, MA, USA) in HEK293T cells. Briefly, cells were passaged and grown at 80–90% confluency in DMEM high glucose media supplemented with 4 mM glutamine, 1 mM sodium pyruvate and 5% heat inactivated FBS. For transfection, reagents were mixed in Opti-MEM (Cat# 31985070, Thermo Scientific) supplemented with 4 mM glutamine and co-incubated for overnight. Next day media was changed with Opti-MEM supplemented with 5% FBS, 4 mM glutamine and kept further for 24 h. Viruses were harvested using 0.45 μm filter and kept in −80 °C for further use.

Upon infection with libraries, we chose the MOI of about 10–20%, so that on average each cell receives only one viral particle. After infection, cells carrying lentiviruses were selected with puromycin and further divided into groups for either injection into mice or drug treatment in culture.

### 2.6. Xenografts

All animals were housed under pathogen-free conditions, and all animal procedures were approved by Ariel University Institutional Animal Care and Use Committee. Cell line suspensions were prepared in 1:1 matrigel (ECM Gel #E1270, Sigma-Aldrich) and approximately 1.5 × 10^6^ cells were subcutaneously injected into the flanks of 9-week-old female nude (Nu/Nu) mice (Envigo, Ness Ziona, Israel). Tumors were measured with calipers and the tumor volume was calculated according to the formula Vol = 0.52 × L × W2. When tumors reached the volume of approximately 200–300 mm^3^, mice were randomly assigned into either treatment group or control group. FX9847 (100 mg/kg) was administered intraperitoneally once daily and osimertinib (2.5 mg/kg) was administered by oral gavage once daily. Tumor volumes were measured, and body weights were monitored twice weekly for the entire span of the experiment.

### 2.7. Genomic DNA Extraction

Two methods of DNA isolation were used following tissue disruption with metal beads using Heartbreaker 2/12. DNA from primary tumor lysate (10 mg of protein) was extracted with proteinase K digestion (Proteinase K, Cat# EO0491, Thermo Scientific) according to the manufacturer’s protocols, followed by phenol/chloroform extraction, and followed by ethanol precipitation repeated twice. Of note, this method is not suitable for DNA isolation from liver and lungs, since chloroform suppresses PCR reactions. Alternatively, for isolation of DNA from primary tumors, liver and lung, we used DNeasy Blood & Tissue Kit (Cat# 69504, Qiagen, Hilden, Germany), followed by additional clean up using DNeasy PowerClean Pro Cleanup Kit (Cat#12997-50, Qiagen) according to the manufacturer’s protocols. Isolation of genomic DNA from cultured cells was performed by Wizard genomic DNA isolation kit (Cat#A1120, Promega, Madison, WI, USA).

### 2.8. Amplification of the Barcodes

We first PCR using a wider separated primer set to a region that covers the entire region of barcodes, and then perform a second PCR reaction with primers that are closer to each other but still cover the barcodes region (Appendix A). Detailed procedure of PCR and primer details described in Appendix A. Briefly, 1-st PCR (PCR 1) was performed using Titanium Taq DNA Polymerase (Cat# 639209, Takara Bio, San Jose, CA, USA) for all above experiments. Separation of the PCR products from primers and gel purification was done by QIAquick PCR & Gel Cleanup Kit (Cat#28506, Qiagen). Afterwards, 2-nd PCR (PCR 2) was carried out using nested primers either generic or having unique sample barcodes as two rounds of nested PCR are necessary to increase the specificity of polymerization. PCR 2 was performed using Phusion High-Fidelity PCR Master Mix (Cat#F531, Thermo Scientific). Since several samples could be sequenced in one chip, all corresponding samples were multiplexed by adding an additional sample barcode during the second round of PCR. Samples were normalized individually, then pooled together, and purification of the PCR products was completed using AmpureXP magnetic beads (Cat#A63882, Beckman Coulter, Brea, CA, USA) following manufacturer’s protocols. Next, we sequenced the barcodes using Ion Torrent and Illumina platforms.

### 2.9. Analysis of the Data

We used a combination of custom-tailored applications to analyze sequencing reads along with the R programming language. Data were first checked for quality of reads through FastQC (v0.11.7), further using barcode-splitter (v0.18.6) reads were demultiplexed based on sample barcodes (1 error as mismatch or deletion was allowed for sample barcodes while demultiplexing). Obtained FASTQ files were used to count the library barcodes by using python-based applications that were custom-made for this purpose. Quantification of the unique barcodes that were abundantly enriched or lost after treatment has been done via a python-based script (software version 3.10.0). For data cleaning and visualizations tidyverse-v1.0.0 and ggplot2-v3.3.3 of R packages were utilized.

### 2.10. Statistical Analysis

We used an R programming language and GraphPad Prism (v9, RRID:SCR_002798) for plotting of data and associated statistics.

## 3. Results

### 3.1. Double-Barcoding System

Previously monitoring cancer development with cell barcoding was done using a barcode library with random barcodes that allows individual labeling of clones within the cancer cell population [6,11,15,17,18]. Since this approach did not allow quantitative comparison of clonal sizes between animals (see Introduction), we realized the necessity of internal controls for normalization. Accordingly, we sought to test effects of a drug on xenograft established by co-injection of drug-sensitive and drug-resistant variants of cancer cells labeled with different barcodes, and use a change in their ratio following the treatment to quantify the effect. Utilizing the ratio between drug-resistant and drug-sensitive cell lines as a measure of drug effect corrects for differences related to biological variations between animals, as well as for DNA quality, since barcodes from both drug-sensitive and drug-resistant cells are PCR-isolated simultaneously from the same total tissue DNA. Moreover, co-injection of different cell populations, e.g., different cell lines or different mutant variants within one cell line, also allows quantitative comparison of their ability to form tumors and metastasis. The common approach to test for drug effects in preclinical studies is subcutaneous cancer cell injection, which is routinely used by CROs and is accepted by FDA (see, for example [1,2,3]). Accordingly, we used this in vivo model, though other models, e.g., PDX or orthotopic models, can also be used.

To simultaneously monitor different cell populations and their clonal structure, we constructed twenty one libraries caring two barcodes. The 8-nucleotide first barcode (Barcode 1) was unique for each library and permitted us to distinguish between various cell lines or cell populations. It was followed by a second barcode (Barcode 2) made of 14 random nucleotides that enabled individually labeling cells within each cell line (Figure 1A). The second barcode allowed quantification of a progeny of each individual cell that forms a primary tumor or metastatic clone, thus providing a measure of a size of each clone or metastasis. The diversity of each library varied between 40,000 and 200,000 of second barcodes. In further experiments, we used libraries with 200,000 barcodes diversity.

### 3.2. Monitoring Tumor Formation and Metastasis by Mixed Population of Cancer Cells

To test the application of the double-barcoding system, we compared the ability of simultaneously injected cancer lines to form primary tumors and metastasis. Accordingly, we barcoded two non-small cell lung cancer (NSCLC) cell lines, A549 and H1975, with two libraries, each having its own barcode 1 followed by random barcode 2. Then 500,000 barcoded cells of each type were mixed together and injected into nude mice to establish xenografts. After formation of tumors, animals were sacrificed and tumors were collected along with lungs and livers. Following DNA isolation and sequencing, barcodes were analyzed. The standard deviation of ratios of H1975 and A549 cells in primary tumors (ratios of the numbers of barcodes 1 that mark H1975 and A549 cells) were around 20%, which is significantly lower than deviations in tumor sizes usually obtained by conventional methods. In metastasis, the standard deviations of these ratios were 30–35%, which is also lower than usually seen deviations between animals in a group by either imaging or direct counting of metastasis.

Co-injection of two lung cancer lines allowed comparative analysis of the ability of cells to establish primary tumors as well as to compare their metastatic capacities. It is noteworthy that, unlike conventional methods, measuring barcodes provides sufficient sensitivity to detect small metastases and evaluate them quantitatively.

Interestingly, the abilities of cells to establish primary tumors and their metastatic capacities did not correlate with each other. Thus, representation of H1975 cells in the primary tumor was about 5-fold higher than A549 (based on barcodes 1 ratio). However, in both liver and lung metastasis, these cells were present in almost equal proportions (Figure 1B). Such comparative analysis of capacities of cells to form primary tumors and metastasis could be especially useful in studies of effects of mutations. Of note, these experiments share the same logic with classical comparative fitness experiments when cell mixing allowed detecting even small differences in effects of mutations on cell growth [25,26]. In fact, cancer cell mixing experiments have been used to compare drug effects in vitro [10,27], but the barcoding in these works was not designed to monitor effects on individual clones, which can significantly enrich the obtained information.

To evaluate the number of metastases formed by A549 cells and their sizes, we analyzed distribution of second clonal barcodes isolated from lungs and liver normalized by the ratio of barcodes 1 (H1975/A549). In line with previous reports [19,28], there was a significant overlap between identical barcodes in these two organs (15%), indicating that if a cell can form metastasis in lungs, it has a high chance forming metastasis in liver (Figure 1C). Furthermore, faster growing metastases were able to better propagate in both lungs and livers (Figure 1D, left panel). Accordingly, evolution of metastatic properties provides advantages for metastasis in both organs. In contrast, there was no correlation between the ability of cells to propagate in primary tumors and in the metastasis that they produce (Figure 1D, right panel).

### 3.3. Using Double Barcoding to Quantify Drug Effects

Next, we investigated the potential of our double-barcoding system to quantitatively assess drug effects on tumor formation and metastasis as compared to conventional measurements. For this analysis we chose a standard drug, osimertinib, an inhibitor of a mutant EGFR (T790M). To evaluate the technology with a novel drug candidate in preclinical stage of development, we used FX9847, a selective inhibitor of DYRK1B kinase. DYRK1B prevents cancer cells from staying in G0 stage of the cell cycle, which is the most chemotherapy-resistant stage. Accordingly, treatment with DYRK1B inhibitors forces cancer cells to enter the cell cycle and makes them drug-sensitive [29,30,31]. Indeed, a combination of osimertinib and FX9847 demonstrated synergistic cell killing in vitro [31,32]. Since information about FX9847 has been published only as conference abstracts, a more detailed description of this drug and its mechanisms of action is presented in the Appendix A.

Results of conventional monitoring by caliper measurements of effects of these two drugs on tumor established by H1975 is shown on Figure 2A. FX9847 alone demonstrates minimal suppression of the tumor growth. On the other hand, osimertinib had a strong tumor-suppressing effect, which was significantly enhanced by a combination with FX9847.

Next, we barcoded osimertinib-sensitive H1975 cells expressing mutated EGFR (T790M) and osimertinib-resistant A549 cells expressing wild type EGFR with two different double-barcode libraries, as in the previous section. We confirmed that H1975 cells are significantly more sensitive to osimertinib than A549 cells in vitro (Figure 2B). Equal numbers of barcoded H1975 and A549 cells were co-injected into nude mice and when tumors became pulpable, animals they were dosed with (a) osimertinib, (b) FX9847, (c) combination of osimertinib and FX9847, or (d) sham was administered. Following treatments, animals were sacrificed, and tumors, lungs and livers were collected. DNA was isolated, and barcodes were amplified and sequenced, as in the previous experiments.

Overall, representation of the cell-line specific barcodes 1 (that marked H1975 and A549 cell lines) indicated that in untreated animals H1975 cells have a higher ability to form tumors than A549 cells (Figure 2C). Treatment with osimertinib strongly reduced the proportion of H1975 cells compared to A549 cells in the primary tumors, reflecting its higher sensitivity to the drug. As negative controls for drug effects, we present effects of osimertinib and FX9847 on the drug-resistant A549 cells in primary tumor and lung metastasis (Appendix A). Interestingly, while FX9847 alone only slightly changed the ratio between H1975 and A549 cells, it significantly enhanced H1975 sensitivity to osimertinib (Figure 2C). These data were in line with the data obtained by caliper measurements of tumor sizes (Figure 2A). Importantly, due to normalization by the internal negative control (i.e., looking for ratios between sensitive H1975 and resistant A549 lines), we were able to obtain statistically significant effects of drugs using only three mice per group.

Comparison of standard- and double-barcoding systems is schematically explained in Figure 3A. To highlight the importance of the internal normalization by the drug-resistant cells, which reflects the difference between standard barcoding and double barcoding, we compared data on the H1975 cells in primary tumors and lung metastasis without and with such a normalization. To evaluate the data reflecting standard barcoding, we present raw numbers of barcode 1 reads (representing the number of H1975 cells) corrected by the number of PCR cycles needed for the barcode amplification. To evaluate the data reflecting double barcoding, we present ratios of barcode 1 reads corresponding to H1975 and A549 cells in the same mice. Figure 3B shows that without normalization, differences between the number of reads for barcode 1 that labels H1975 cells in the tumors in different animals within a group was orders of magnitude, which made the comparison of control and osimertinib-treated groups practically useless. On the other hand, upon normalization by A549 cells, the number of reads within the groups became much closer, and the effect of osimertinib became clearly evident (Figure 3B). This comparison highlights the fundamental advantage of double barcoding over a standard cell-barcoding approach in measurement drug effects. 

Additional advantage is achieved through utilization of the random barcode 2 in each library, as it allows quantitative assessment of clonal sizes in primary tumors and metastasis based on the numbers of reads of each clone-specific barcode 2. The number of reads representing each barcode 2 within a population of cells is proportional to the number of cells in the corresponding clone produced by a progeny of one cell (and therefore each metastasis). Normalization of numbers of barcodes 2 reads on the ratios of cell line-specific barcodes 1 (H1975/A549) allows comparison of the clonal sizes in different animals, which provides significant advantages over standard barcoding in measuring drug effects. Furthermore, with some drugs that similarly suppress all clones in the population (see below), only double barcoding can provide data on drug effects, while standard barcoding cannot be used at all.

### 3.4. Clonal Responses to Drugs

Individual cell barcoding and monitoring clonal fates allowed us to uncover an unexpected feature of a drug action. We normalized random barcodes 2 by the ratio of barcodes 1, as in previous experiments, and plotted the distribution of barcodes 2 by the number of reads, which reflects sizes of individual clones within the tumor. Figure 4A shows a typical distribution curve in untreated tumors. Such curves were highly reproducible between animals within each group (Appendix A). Treatment with FX9847 alone did not change either the shape of the curve or its position, confirming minimal effect of FX9847. In contrast, osimertinib strongly shifted down the entire curve without significantly changing its shape (Figure 4A). This finding indicates that osimertinib equally suppresses growth or kills cells derived from different clones. In other words, there was little clonal variability in sensitivity to osimertinib in cells that formed primary tumor.

The combination of osimertinib and FX9847, while reducing the ratio of H1975 and A549 cell populations (i.e., ratio of the numbers of corresponding barcodes 1) only 2-fold, compared to osimertinib alone (Figure 2C) dramatically changed the shape of the curve of distribution of clones by size (i.e., number of reads per individual barcode 2) (Figure 4A). In fact, the drug combination reduced the number of clones that survived the treatment almost 5-fold, compared to osimertinib alone. 

These data indicate that adding FX9847 to osimertinib did not equally suppress the growth of all clones, but rather eliminated a large number of clones while sparing the others. In other words, FX9847 had a synergistic effect with osimertinib against most clones except the few that were resistant to the combination. Thus, this analysis allowed us to quantitatively assess the internal heterogeneity of tumor-forming cells. Overall, these experiments uncovered two fundamentally different effects of drugs on cancer cell populations: the EGFR (T790M) inhibitor alone kills cells independently of their clonal properties, while in the presence of the DYRK1B inhibitor it kills cells selectively, depending on their clonal origin. Thus, our system provides a unique opportunity to detect potential clonal-dependent resistance mechanisms in the tumor cell population (see below).

Compared to primary tumors, in metastasis the effects of osimertinib were much weaker (Figure 4B,C), suggesting that metastasis acquired partial resistance to the drug. Nevertheless, the combination of osimertinib and FX9847 significantly reduced the number and sizes of metastasis in lungs, while either drug alone did not show significant effects (Figure 4C). Importantly, as with the primary tumors, in metastasis the effects of FX9847 were clonal-specific, while the effects of osimertinib were clonal-independent (Figure 4B).

### 3.5. Evaluating Clonal Effects of Other Drugs

To understand how general differences in clonal effects of drugs are, we tested an unrelated set of drugs, including a topoisomerase 1 inhibitor, irinotecan, and an inhibitor of DNA methylation, 5-azacytidine. Since these drugs are used for treatment of colon cancer, we tested their effects on a colon cancer cell line, HCT116. We titrated irinotecan and identified IC50, which appeared to be 20 nM. The 5-azacytidine alone did not kill these cells even at high concentrations, but it significantly sensitized cells to irinotecan (not shown). In the initial experiment, cells were barcoded and exposed to irinotecan. Distribution of barcodes demonstrates that irinotecan both shifts the curve down and changes its shape (Figure 5A), suggesting that it has stronger killing effects on some clonal subpopulations. 

To test whether genetic differences are responsible for clonal effects of irinotecan, we cloned the parental population of HCT116 cells to obtain genetically identical clonal populations. One of the clones was barcoded and exposed to irinotecan. With such cloned cells, irinotecan only shifted the barcode distribution curve down without changing its shape (Figure 5B). We concluded that with a genetically uniform cell population, irinotecan suppresses growth or kills cell population uniformly. We tested irinotecan sensitivity of a few additional clones obtained from the original cell line and found that they show almost 20-fold variability in IC50 towards irinotecan, ranging from 2 to 40 nM. Therefore, the original clonal effects of irinotecan were associated with the genetic heterogeneity of the cell line, and these effects were diminished in the genetically uniform population. 

On the other hand, the effects of 5-azacytidine on HCT116 cells were very different. In the heterogeneous population, 5-azacytidine added in combination with irinotecan, and both shifted the size distribution curve and changed their shape much more than with irinotecan alone (Figure 5A). When the effects of 5-azacytidine were tested in the genetically homogeneous population of cells, we observed that the addition of 5-azacytidine together with irinotecan changed the shape of the curve even more dramatically, demonstrating that this drug effectively eliminates a subset of sub-clones without touching the rest of the population (Figure 5B). Therefore, here again we encountered two different types of drugs with different clonal effects. 

## 4. Discussion

Here, we introduce a novel barcoding system that allows monitoring the fate of individual clones in multiple cell populations to quantitatively monitor cancer cell behavior in animals in basic research and preclinical studies. Currently used approaches that involve imaging or direct measurements suffer from relatively low sensitivity and low resolution, leading to the inability to quantitatively evaluate the number of metastases and the sizes of individual metastasis. There have been reports of measurements of metastasis by standard cell barcoding, but this system has significant limitations. As noted above, standard cell barcoding allows assessment of drug effects on the number of clones (i.e., the number of recovered barcodes) that formed primary tumors and metastasis [23,33], but not on the clonal sizes. However, with some drugs the number of barcodes cannot be used for assessing drug effects. For example, we detected only statistically insignificant differences in the number of barcodes between control and osimertinib-treated mice (see Figure 3C) despite the strong tumor-suppressing effect of the drug (see Figure 2C). This is because osimertinib suppresses the growth of all clones without selectively killing a subpopulation of clones. Therefore, with such drugs, regular barcoding cannot be used at all for assessing drug effects, and with other drugs only part of the drug effects can be seen. In contrast, double barcoding that allows normalization clearly uncovers drug effects.

An important advantage of the double-barcoding system over single barcoding with random barcodes is that it allows internal normalization of the results, which strongly enhances statistical significance of the results. The deviation in measurements of cancer parameters and drug responses between animals within a group are of two different origins: (1) biological difference between animals (e.g., difference in their epigenetic nature), and (2) technical differences in the process of isolation of DNA from primary tumors and distant organs. Normalization with the double-barcoding system allows correction of both sources of deviations. Indeed, when we inject both drug-sensitive and drug-resistant cell lines in a mouse, both experience similar host effects on tumor development and metastasis, and DNA from both lines is isolated with the same impurities. When we compared deviation from average with and without normalization by the second cell line, we observed that the normalization strongly reduces the deviations (Figure 3B), thus allowing the use of a relatively small number of animals per group to achieve the statistical significance. With this level of precision, three–four animals per group should be sufficient for almost any study of drug effects.

In order to monitor drug effects by the double-barcoding system, drug-resistant and drug-sensitive versions should be co-injected. Drug-resistant versions could be selected from the drug-sensitive population by standard approaches, i.e., dose escalation selection [34], which will work best for targeted drugs. Alternatively, if one deals with a small molecule drug, even with not-targeted action, overexpression of an MDR pump may make cells drug-resistant [35]. When studying immunotherapeutic drugs, one can suppress generation of the peptide-loaded MHC1, e.g., by depletion of TAP transporters, which will make cells resistant to most types of immunotherapy [36,37]. On another note, as mentioned above, within standard cell lines, individual clones demonstrate entirely different drug sensitivities (e.g., different clones of HCT116 cells had a 20-fold difference in sensitivity to irinotecan), and simple cell cloning can generate both drug-sensitive and drug-resistant versions of the same cell line. 

Effects of drugs measured by the barcode sequencing were apparently stronger than effects measured by caliper. In fact, when measured by caliper, by day seven of exposure we observed about a 4-fold difference in tumor sizes between control and osimertinib-treated animals. However, when measured by the barcode sequencing, we observed a 12-fold difference. We suggest that an important source of the variability upon measurements of primary tumor volumes with caliper is that it does not consider tumor necrosis, which could lead to significant underestimation of drug effects. However, measurements by barcoding are likely to detect only live tumor cells, which avoids counting necrotic cells and thus underestimation of drug effects. Thus, estimation of the drug effects with the barcoding system is significantly more accurate than with conventional methods. 

Plotting the barcodes according to their presence in tumors or metastasis, i.e., size distribution of the barcoded clones, revealed novel and unexpected drug properties. It appears that some drugs, e.g., osimertinib, reduce the number of cancer cells in the population independently of their clonal origin, while other drugs, e.g., FX9847, kill some cancer clones and do not affect others. Finding that some drugs suppress certain clones but not the others, while other drugs suppress the entire clonal population, could be critical for designing treatment strategies and finding novel drug combinations. In fact, if there is a strong clonal selectivity in response to a drug, dose escalation of this drug does not make much sense and could be even harmful since resistant clones will survive slightly higher doses. In contrast, for drugs that affect the entire population of cancer cells, dose escalation could be beneficial. Overall, our newly developed double-barcoding system expands our capabilities in studies of tumor development and metastatic behavior of cancer cells and drug responses.

### Potential Sources of Errors in Random Barcoding

A cell receiving more than one viral particle and thus two barcodes will be counted as two different clones. Accordingly, in studies of drug effects, the number of clones will be overestimated. To alleviate this problem, low viral MOI is used to make sure that most cells receive no more than one barcode. Since infection events are independent on each other, in our experiments at 20% infection rate, after selection 100% of remaining cells carry at least one barcode, and 20% of these cells are infected with two viruses, and therefore have two barcodes. Accordingly, this effect generates a 20% error in quantifying the number of metastatic clones. If such an error is problematic, one can reduce the MOI upon infection. At an infection rate of 5%, the error becomes practically negligible. Furthermore, knowing the infection rate, one can simply correct for such an error in final calculations. 

Viruses with the same barcode 2 infect two different cells. Such a scenario will create an opposite problem, i.e., two cells will be counted as the same clone, and thus the number of clones will be underestimated. This is not a problem when drugs that similarly affect all clones are tested. However, if the effects of a drug are dependent on the clonal origin of cells, counting two clones as one may blunt the drug effect. In our experiments, the library diversity was 200,000 and the number of originally infected cells around 100,000 (we infected 500,000 cells with a 10–20% infection rate, selected and allowed to propagate for a couple of generations before injection into animals). Under these conditions, the fraction of same barcodes ending up in two different cells was 27% (according to equation nb/nc×e^(−(nb)/nc) where nb is a number of barcodes in the library and nc is a number of cells). If such an error is problematic, one can increase the diversity of the library. In fact, increasing the diversity twice will reduce such an error to below 7%.

## 5. Conclusions

We developed a modification of the barcoding approach for monitoring tumors and metastasis that is quantitative, highly sensitive and highly reproducible. This system allows comparing effects of drugs on different cell populations and thus normalizing drug effects by drug-resistant lines, which significantly increases the reproducibility of the result. Using this barcoding system, we established that anti-cancer effects of certain drugs are clone-dependent, while other drugs demonstrate clone-independent effects on cancer cell populations.

## Figures and Tables

**Figure 1 cancers-14-01381-f001:**
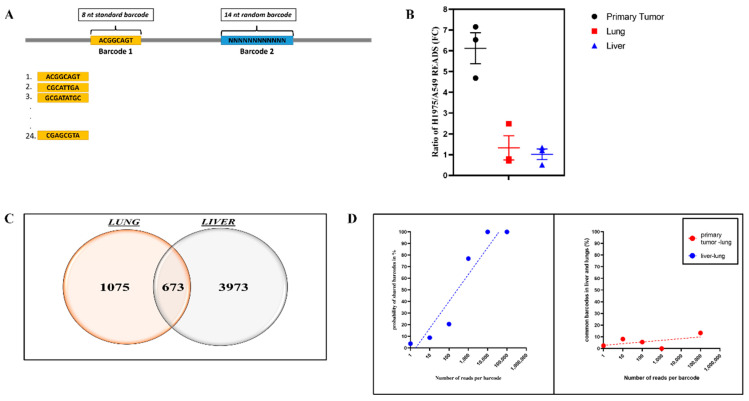
Double-cell barcoding system allows monitoring primary tumors and metastasis derived from several cell lines simultaneously. (**A**) The schematic representation of the double-barcoding system. (**B**) Metastatic ability of H1975 and A549 cell lines. The data reflect the fold change difference of the number of reads corresponding to H1975 and A549 cell lines in primary tumors and metastasis. (**C**) A significant fraction of clones that form metastasis in lungs can also form metastasis in liver. The number of barcodes common and unique for lungs and liver is shown. (**D**) The ability to form metastasis does not correlate with the growth of cells in primary tumors (right panel). However, the ability of clones to populate and grow in lungs closely correlates with the rate of growth in liver (left panel).

**Figure 2 cancers-14-01381-f002:**
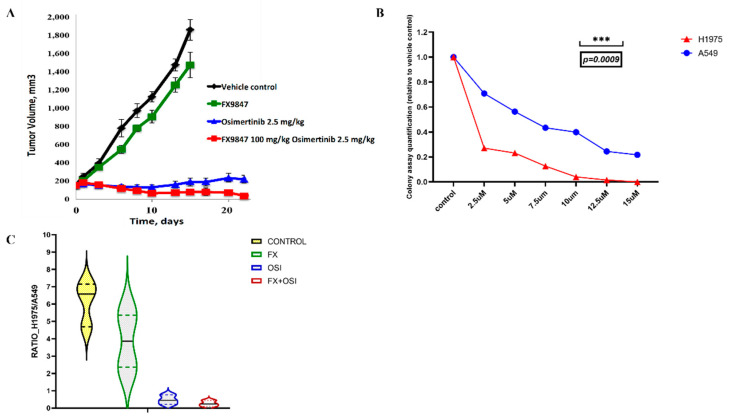
Effects of drugs combination treatments on primary tumors. (**A**) Effects of osimertinib and FX9847 and their combination on xenograft tumor growth measured by the caliper. Xenografts were established with H1975 lung cancer cells, and mice (10 animals per group) were treated with either osimertinib or FX9847 or their combination as described in the Materials and Methods section. (**B**) Sensitivity of H1975 and A549 cells to osimertinib (7 days of treatment). Cell viability was measured by the colony growth assay. (**C**) Effects of osimertinib and FX9847 and their combination on the primary tumors. The number of reads corresponding to H1975 cells was normalized by the number of reads that represent A549 cells. ***, *p* < 0.005.

**Figure 3 cancers-14-01381-f003:**
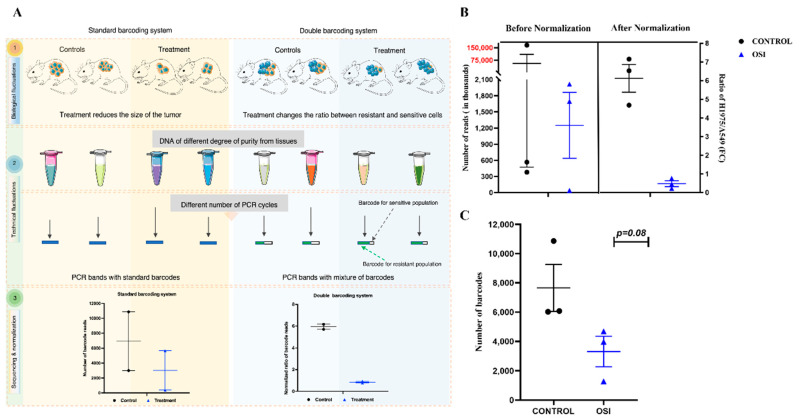
Comparison of standard and double-barcoding systems. (**A**) Schematic explanation of the standard- and double-barcoding systems. There are two sources of variability in counting barcodes. They include biological variabilities associated with differences in cell injections and their ability to form tumors and metastasis in different animals; and technical variability associated with differences between different samples in tissue disruption and DNA isolation, which significantly affect the efficiency of isolation of barcodes for sequencing. (Left panel) shows experiment with standard barcoding. Barcoded cells form tumors and metastasis in control group with certain variability, and these tumors and metastasis might be smaller (and fewer in case of mets) in drug-treated animals. DNA isolated from these animals will be of different quality and different number of PCR cycles will be necessary to amplify sufficient for sequencing amounts of barcode DNA bands (visible bands are needed). Correcting the number of reads after sequencing for tumor sizes and the number of PCR cycles creates enormous variabilities, which makes comparison of control and drug-treated groups almost impossible. Similar is true with metastasis. The only reasonable comparison could be done with the number of clones (number of different barcodes), but this parameter is insufficient for evaluation of drug effects (see the example with osimertinib below). (Right panel) shows experiment with double barcoding. Drug-sensitive and drug-resistant cells are barcoded by two different double-barcoding libraries and co-injected into animals in equal amounts. Both cell populations form mixed tumors and both metastasize to distinct organs. Due to the difference in animals, there is certain variability in tumor sizes, but the relative representation of two cell types in the tumor has low variability. The same is true for metastasis. In the drug-treated group the sizes of tumors do not become smaller because of the presence of the drug-resistant cells. They expand, while the drug-sensitive cells contact. The ratio between the lines changes, and such a ratio also has low variability. DNA isolated from these animals will be of different quality, and different number of PCR cycles will be necessary to amplify sufficient for sequencing amounts of barcode DNA bands. However, bands containing barcodes for both lines are amplified together from the same DNA, and therefore their ratio remains unchanged independently on the number of PCR cycles. Similar is true with metastasis. Therefore, double barcoding diminishes both biological and technical variations and allows quantification not only of the number of clones but also of their sizes. (**B**) Comparison of raw numbers of reads (of barcode 1, reflecting the number of H1975 cells in tumors) in mice treated and untreated with osimertinib with normalized number of reads in the same samples. (**C**) Comparison of the number of clones (barcodes 2) in control and osimertinib-treated group. The experiment shows that upon using standard barcoding without normalization on drug-resistant cells, quantification of effects of some drugs becomes practically impossible.

**Figure 4 cancers-14-01381-f004:**
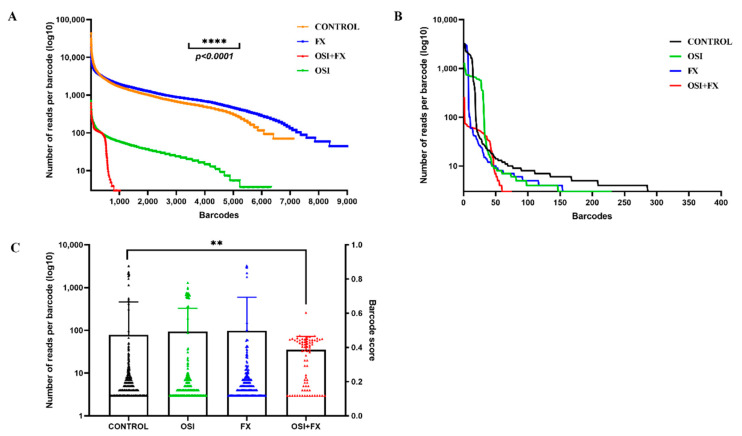
Effects of drug treatments on the size distribution of clones in primary tumors and metastasis. (**A**) Size distribution of clones in primary tumors of control population and animals treated with osimertinib, FX9847 and their combination. (**B**) Size distribution of clones in lung metastasis of control population and animals treated with osimertinib, FX9847 and their combination. (**C**) Quantification of significance of effects represented in Figure 4B. **, *p* < 0.01; ****, *p* < 0.001.

**Figure 5 cancers-14-01381-f005:**
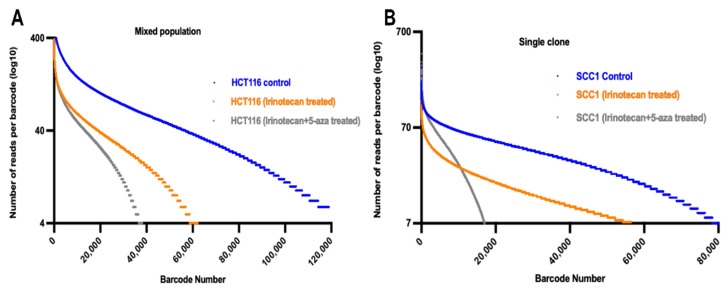
Effects of irinotecan alone or in combination with 5-Azacytidin on clonal size distribution. (**A**) Size distribution of clones of heterogenous cell population either untreated or treated irinotecan or irinotecan + 5-azacytidin. (**B**) Size distribution of clones of homogenous cloned cell population either untreated or treated irinotecan or irinotecan + 5-azacytidin.

## Data Availability

All the data that is critical for the conclusion are present in the manuscript or Appendix A. Any additional requirement of custom codes used could be availed upon request.

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
