# Peer review of "A Cell Double-Barcoding System for Quantitative Evaluation of Primary Tumors and Metastasis in Animals That Uncovers Clonal-Specific Anti-Cancer Drug Effects"

_cancers, 2022, doi:10.3390/cancers14061381_

Round 1

Reviewer 1 Report

This paper presents a very useful methodology for in vivo measurements of metastasis and drug effects on primary and metastatic tumors in a way that overcomes tumor growth variability, allowing for the quantitation of drug effects in a much smaller that usual number of mice. The methodology also allows one to evaluate the clonality of drug resistance in a cell population, which should be especially useful for PDX studies. While the methodology is novel, useful and the results are clear-cut, the MS should be significantly revised to clarify the presentation and to explain the methodology better.

  1. The principal novelty of this method appears to lie not so much in barcoding (a widely used approach) but in simultaneous injection of drug-sensitive and drug-resistant cells, using the barcode-based yield of the resistant cells as the normalization standard for the effects of the drug on the sensitive cells. This approach should work primarily for targeted drugs, which clearly distinguish between sensitive and resistant cell lines but it would not be applicable to all drugs. This should be clearly explained.
  2. The paper states repeatedly that “effects of a novel DYRK1B kinase inhibitor FX9847 on primary tumors and metastasis is clone‐dependent, while a distinct drug osimertinib demonstrated clone‐independent effects on cancer cell populations.” However, it seems that the first part of this statement relates not to the effects of FX9847 but rather to the combination of FX9847 and osimertinib. It seems impossible to tell whether the observed clonal variability reflects the response to FX9847 or to the effects of osimertinib under the conditions of DYRK1B inhibition.
  3. Figure 2:
    1. It would seem more logical to present in vitro data first (Fig. 2B) and then the in vivo results (Fig. 2A and 2C).
    2. 2A: are the differences in tumor volumes between osimertinib and combo statistically significant at any time points?
    3. Were mouse body weights measured? For a drug combination study, I would expect to see a representation of the effects on mouse body weights.
  4. The paper requires significant proofreading, there are quite a few typos and misspellings (e.g. “xenographs”).

Author Response

Comments and Suggestions for Authors 

This paper presents a very useful methodology for in vivo measurements of metastasis and drug effects on primary and metastatic tumors in a way that overcomes tumor growth variability, allowing for the quantitation of drug effects in a much smaller that usual number of mice. The methodology also allows one to evaluate the clonality of drug resistance in a cell population, which should be especially useful for PDX studies. While the methodology is novel, useful and the results are clear-cut, the MS should be significantly revised to clarify the presentation and to explain the methodology better. 

  1. The principal novelty of this method appears to lie not so much in barcoding (a widely used approach) but in simultaneous injection of drug-sensitive and drug-resistant cells, using the barcode-based yield of the resistant cells as the normalization standard for the effects of the drug on the sensitive cells. This approach should work primarily for targeted drugs, which clearly distinguish between sensitive and resistant cell lines but it would not be applicable to all drugs. This should be clearly explained. 

We believe that the approach could be effectively used for both targeted and not-targeted drugs. We emphasize these possibilities in the Discussion: “In order to monitor drug effects by the double barcoding system, drug resistant and drug sensitive versions should be co-injected. Drug resistant versions could be selected from the drug-sensitive population by standard approaches, i.e. dose escalation selection, which will work best for targeted drugs. Alternatively, if one deals with a small molecule drug, even with not-targeted action, overexpression of a MDR pump may make cells drug-resistant. When studying immunotherapeutic drugs, one can suppress generation of the peptide-loaded MHC1, e.g. by depletion of TAP transporters, which will make cells resistant to most types of immunotherapy.  On another note, as mentioned above, within standard cell lines, individual clones demonstrate entirely different drug sensitivities (e.g. different clones of HCT116 cells had 20-fold difference in sensitivity to irinotecan), and simple cell cloning can generate both drug-sensitive and drug-resistant versions of the same cell line.” 

  1. The paper states repeatedly that “effects of a novel DYRK1B kinase inhibitor FX9847 on primary tumors and metastasis is clone‐dependent, while a distinct drug osimertinib demonstrated clone‐independent effects on cancer cell populations.” However, it seems that the first part of this statement relates not to the effects of FX9847 but rather to the combination of FX9847 and osimertinib. It seems impossible to tell whether the observed clonal variability reflects the response to FX9847 or to the effects of osimertinib under the conditions of DYRK1B inhibition. 

We agree with this correction and modified the text accordingly: “Overall, these experiments uncovered two fundamentally different effects of drugs on cancer cell populations, the EGFR (T790M) inhibitor alone kills cells independently of their clonal properties, while in the presence of the DYRK1B inhibitor, it kills cells selectively, depending of their clonal origin.” 

  1. Figure 2: 
    1. It would seem more logical to present in vitro data first (Fig. 2B) and then the in vivo results (Fig. 2A and 2C). 

In Fig. 2A the experiment with xenografts was done with H1975 cells only, not with the mixture of two cell lines. If we added the drug-resistant cells to the xenograft, we would not be able to detect suppression of the tumor growth by the drug because the resistant line would continue growing. Therefore, we believe these data should precede comparison of drug response of the mixture of two lines. Their mixture was done in experiments described in Fig. 2C. 

  1. 2A: are the differences in tumor volumes between osimertinib and combo statistically significant at any time points? 

Though the trend is seen with several time points, the significance is reached only with the last time point where osimertinib alone treated tumors continues to growth, while osimertinib+FX9847 shrinks. 

  1. Were mouse body weights measured? For a drug combination study, I would expect to see a representation of the effects on mouse body weights. 

Unfortunately, body weights were not measured in these experiments.

  1. The paper requires significant proofreading, there are quite a few typos and misspellings (e.g. “xenographs”). 

As requested we performed the proofreading and corrected typos. 

Reviewer 2 Report

In this manuscript, Arkadi Hesin reported a double barcoding system as a cancer cell tracking strategy in evaluating the drug sensitivity among different cancer subpopulations in primary tumor and metastasis in vivo. The main novelty of this design is to introduce a second 12-nt random barcode diversifying as many as up to 200,000 cell individuals. Also, barcode 1 which follows traditional labels helps track cells of the same type, facilitating the normalization of barcode 2 in handling specific cell clone size comparison across different cell types. It’s a good methodology manuscript and my main concern falls in the validation step.

  1. It’s mentioned on line 123, “we chose the MOI of about 10-20%, so that on average each cell receives only one viral particle”. It’s well acknowledged that each cell receives one version of barcode 2 is the ideal scenario and it usually does not happen. So, I believe there will be one single cell receives 2, 3 or even more particles while some may not even get one. This does not affect the barcode 1 mediated quantification but it definitely deteriorates the efficacy of barcode 2’s. To push the limit and exemplify, if three H1975 cells (A, B and C) luckily get a common barcode 2 (a) as well as a different second one (b, c and d), so the resultant cell labeling would be A-ab, B-ac and D-ad. If cell A carrying barcode a and b undergoes lung metastasis but only B and D cells present in lung metastasis, the conclusion from Figure 1C may be challenged since barcode 2 of “a” that’s found in both cells A and (B,C) does not belong to same cell clones. Therefore, it’s crucial to do a simple validation step from in-vitro cultured cells post transfection.

Author Response

Comments and Suggestions for Authors 

In this manuscript, Arkadi Hesin reported a double barcoding system as a cancer cell tracking strategy in evaluating the drug sensitivity among different cancer subpopulations in primary tumor and metastasis in vivo. The main novelty of this design is to introduce a second 12-nt random barcode diversifying as many as up to 200,000 cell individuals. Also, barcode 1 which follows traditional labels helps track cells of the same type, facilitating the normalization of barcode 2 in handling specific cell clone size comparison across different cell types. It’s a good methodology manuscript and my main concern falls in the validation step. 

  1. It’s mentioned on line 123, “we chose the MOI of about 10-20%, so that on average each cell receives only one viral particle”. It’s well acknowledged that each cell receives one version of barcode 2 is the ideal scenario and it usually does not happen. So, I believe there will be one single cell receives 2, 3 or even more particles while some may not even get one. This does not affect the barcode 1 mediated quantification but it definitely deteriorates the efficacy of barcode 2’s. To push the limit and exemplify, if three H1975 cells (A, B and C) luckily get a common barcode 2 (a) as well as a different second one (b, c and d), so the resultant cell labeling would be A-ab, B-ac and D-ad. If cell A carrying barcode a and b undergoes lung metastasis but only B and D cells present in lung metastasis, the conclusion from Figure 1C may be challenged since barcode 2 of “a” that’s found in both cells A and (B,C) does not belong to same cell clones. Therefore, it’s crucial to do a simple validation step from in-vitro cultured cells post transfection. 

We apologize, but the question raised by the reviewer was not clear to us. Especially, we are not sure about the example with H1975 A, B and C cells. Accordingly, we need some extra explanation of what needs to be validated if our response below is insufficient.

Our understanding is that the reviewer is concerned that some cells can receive more than one viral particle and therefore more than one barcode. Upon even 20% infection rate, after the puromycin selection, only cells with barcodes will survive. Since infection events are independent on each other, 20% of this population will be infected with two viruses, and therefore have two barcodes. (The percentage of cells with three barcodes will be 20% X 20% = 4%). If a cell that carries two barcodes forms metastasis, this clone will be perceived in our quantification as two clones. Accordingly, this effect will generate 20% error in quantifying the number of metastatic clones. The contribution of cells that carry three barcodes in such an error is much smaller. If the infection rate is 10%, rather than 20%, the error will be proportionally smaller. For all practical purposes, such an error should not be a problem. However, if the error is problematic, one can reduce the MOI upon infection to make the error negligible.  Furthermore, knowing the infection rate, one can simply correct for such an error in final calculations. Moreover, such errors are similar in control and drug-treated animals since the same barcoded cell culture is injected in all groups of animals.  We introduced these considerations in Discussion.

Round 2

Reviewer 1 Report

The revision fully addressed my concerns.

Author Response

Thank you for your time

Reviewer 2 Report

In author’s responses, they find it unclear of my question and I’m happy to rephrase here. In brief, my concern is about whether or how much the extent the random barcode 2 can represent each cell clone in the bulk sequencing analysis. If any cell receives more than one viral particle or barcode 2, then this cell clone is not unique to the barcode 2 since other cells may get the same barcode 2 and it’s beyond the capability of bulk DNA sequencing to tell which cell clone is metastatic even if barcode 2 reading was found in both liver metastasis and the primary tumor.

Also, the author may misuse the concept of MOI and infection rate. As the author puts it, “we chose the MOI of about 10-20%, so that on average each cell receives only one viral particle” and puromycin was used to select transduced cells. However, puromycin only kills cells without any virus transduced but keeps all cells with either one or multiple barcodes. In addition, “If a cell that carries two barcodes from metastasis, this clone will be perceived in our quantification as two clones”. That doesn’t solve the problem since the author are trying to equalize “each cell clone” and “each unique barcode 2”, which is not the case due to the bulk DNA sequencing not being able to tell individual cell clones and necessary validation step was eliminated in this manuscript.

Author Response

We appreciate the reviewer's comments, and now we understand potential problem with two cells receiving the same barcode 2 that previously we have not recognized. We found how the fraction of such cells can be calculated based on diversity of the library and the number of infected cells and calculated such error for our experiments.  Overall, we recognize two sources of errors, the first one related to one cell receiving two barcodes, which will overestimate the number of clones, and the second one related to a barcode going into two different cells, which will underestimate the number of clones. We put these explanations in a separate sub-chapter of Discussion and suggested ways to reduce these errors, if necessary.

Round 3

Reviewer 2 Report

In this revision, I'm happy to see the added subsection in the discussion explicitly bringing up the limitation of this study which is very important if other researchers in this field are to apply this method in different studies. Though this revise does not increase the data quality, it does increase the scientific soundness.